# Effects of Chromium Toxicity on Physiological Performance and Nutrient Uptake in Two Grapevine Cultivars (*Vitis vinifera* L.) Growing on Own Roots or Grafted onto Different Rootstocks

Kleopatra-Eleni Nikolaou [1,*], Theocharis Chatzistathis [2], Serafeim Theocharis [1], Anagnostis Argiriou [3], Stefanos Koundouras [1] and Elefteria Zioziou [1]

[1] School of Agriculture, Aristotle University of Thessaloniki, 54124 Thessaloniki, Greece; sertheo@agro.auth.gr (S.T.); skoundou@agro.auth.gr (S.K.); ezioziou@agro.auth.gr (E.Z.)
[2] Hellenic Agricultural Organization (H.A.O.) 'Demeter', Institute of Soil and Water Resources, 57001 Thessaloniki, Greece; t.chatzistathis@swri.gr
[3] Institute of Applied Bioscience 6th Km Charilaou–Thermi Road, 60361 Thessaloniki, Greece; argiriou@certh.gr
[*] Correspondence: nikolaouk@agro.auth.gr

**Abstract:** Chromium toxicity is considered within the most severe and dangerous nutritional disorders, and it can often be observed in crops grown in industrial areas. The present study aims to determine the effects of Cr(VI) toxicity on the growth, nutrition, and physiological performance of grapevines. In a pot hydroponic experiment, own-rooted Merlot and Cabernet Franc grapevine cultivars or cultivars grafted onto 1103P and 101-14 Mgt rootstocks were exposed to 120 μM Cr(VI). Leaf interveinal chlorosis appeared after forty-five days of treatment. Overall leaf chlorosis and brown root coloration after sixty days was reported. A significant effect on the majority of the measured parameters due to the Cr(VI) treatment was observed. Chromium stress increased the total Cr concentrations in all parts of the vines, i.e., leaves, shoots, roots, and trunks. When comparing between the studied plant sections, the roots presented the highest Cr concentrations, ranging from 396 to 868 mg kg$^{-1}$ d. w., and then, in descending order, the Cr concentrations ranged from 41 to 102 mg kg$^{-1}$ d. w. in the trunks, from 2.0 to 3.3 mg kg$^{-1}$ d. w. in the leaves, and from 1.9 to 3.0 mg kg$^{-1}$ d. w. in the shoots. Between the assessed rootstocks, 1103P was identified to be a better excluder of Cr concentration in the roots and other aerial parts of the vines. Additionally, chromium toxicity negatively affected the concentrations and compartmentalization of the most important nutrients. Leaf chlorophyll (Chl) concentration decreased down to approximately 53% after sixty days of Cr stress. Chromium toxicity significantly reduced the stem water potential (SWP), net $CO_2$ assimilation rate (A), stomatal conductance (gs), and PSII maximum quantum yield in all the cases of grafted or own-rooted vines. At this stage, chromium stress increased the leaf total phenolic content from 46.14% in Merlot vines to 75.91% in Cabernet Franc vines.

**Keywords:** grapevine; chromium stress; rootstocks; ion concentration; $CO_2$ assimilation; stomatal conductance; chlorophyll fluorescence

## 1. Introduction

Chromium (Cr) is typically found in rocks, soil, and groundwater and is highly toxic for humans, animals, plants, and microorganisms [1]. Cr commonly appears in nature in two oxidation states: Cr(III) and Cr(VI). The latter, Cr(VI), has been identified as toxic and can cause various types of cancer or DNA damage. Thus, an increased, focused attention on this metal has been drawn from the scientific community [2]. The impact of Cr(VI) toxicity in plants can be observed at multiple levels, such as growth and biomass reduction, negative effects on leaf and root growth, or inhibition of enzymatic activities and mutagenesis [3]. Cr(VI) toxic effects are also evident, showing symptoms such as delay in seed germination, damaged roots and reduction of root growth, reduced biomass,

reduction in plant height, photosynthetic impairment, membrane damage, leaf chlorosis and necrosis, and eventually plant death [4]. Each year, huge amounts of Cr(VI) are released into the environment from natural sources, as well as industry, mining, and agricultural activities [5]. Regarding Cr's natural origin, mineral leaching is considered the main cause of chromium occurrence in groundwater. The natural occurrence of chromium is verified by the presence of Cr-containing minerals (chromite, Fe-chromite, Cr-bearing goethite, and silicates). The contact of water with ultramafic rocks and soils, such as serpentines, dunites, and ophiolites, has also been identified as a cause of high Cr(VI) concentrations in numerous cases [6–8]. In addition, Cr(VI) is released into the environment from a wide range of industrial activities, namely electroplating, cement plants, steel production works, leather and wood preservation, metal finishing and metal plating, timber processing, pigmentation, paint and paper production, tobacco smoke, and leaching from improper sanitary landfills [9,10].

In Greece, extensive contamination of the surface and groundwater with chromium was found in the Asopos river [11]. In detail, the Asopos springs from the northern slope of the Cithaeron Mountain southwest of Thebes (approximately 85 Km north of Athens) and empties into the South Euboean Gulf. According to other studies [12], a wide spatial variability of the total chromium content in the groundwaters of the Asopos basin was found, reaching up to 180 μg/L. Additionally, the presence of significant Cr(VI) concentrations in drinking water sources was detected in several locations in Greece [13].

Grapevines (*Vitis vinifera* L.), a member of the *Vitaceae* family, are one of the most important fruit crop species, mainly used for grape production. They grow, in most cases, grafted onto different rootstocks because of phylloxera, a root-feeding aphid that is capable of damaging grapevine roots and destroying vine plants of the species *Vitis vinifera* L. American vine species are used as rootstocks to prevent the severe damage of root systems caused by phylloxera. Cr, in contrast to other toxic trace metals, has received little attention from grapevine scientists. Although several studies have been documented on the toxic effects of Cr in different cultivated plant species, there are very few reports on grapevines [14,15]. Elevated concentrations of this heavy metal in human diets constitute a potential health hazard when it enters the food chain. Food crops represent an important pathway for the movement of potentially toxic metals from soil and water to human beings [16,17]. Grapes of *V. vinifera* L. are used for both table grape and wine consumption. Furthermore, vine leaves compose a common ingredient of the Mediterranean diet. Stuffed grapevine leaves is a traditional eastern Mediterranean course made of grapevine leaves.

The present study investigates the effects of Cr(VI) toxicity on the growth and physiological performance of the grapevine. In addition, nutritional changes and Cr accumulation in different parts of the vines are also determined. For this purpose, in a hydroponic vine-growing experiment, two grapevine cultivars (Merlot and Cabernet Franc) grafted onto two different rootstocks (1103P and 101-14 Mgt) are evaluated.

## 2. Materials and Methods

### 2.1. Plant Material and Experimental Conditions

The present pot culture study was conducted from April to September 2020 at the experimental farm of Aristotle University of Thessaloniki, which is located 15 Km southeast from the Thessaloniki city residential area and, geographically, lies at a latitude of 40°32.267′ and a longitude of 22°59.885′.

Two-year-old own-rooted vines of Merlot and Cabernet Franc cultivars or vines grafted onto 1103 Paulsen or 101-14 Mgt rootstocks that were pruned to two winter buds with roots of 8 to 9 cm in length were planted in 10 L plastic pots filled with a 1:1 sand:perlite mixture and placed outdoors. The experiment was a completely randomized factorial design consisting of 3 (grafted on two rootstocks or own-rooted vines) × 2 (cultivars) × 2 (presence or absence of Cr(VI)) with six uniform potted vines per cultivar or cultivar—rootstock combination. After the initial establishment, the vines were irrigated automatically every two days using a drip irrigation system with a 650 mL capacity per plant of modified

50% Hoagland's No. 2 nutrient solution [18]. In Cr(VI)-treated plots, the vines received 650 mL chromium solution (120 μM Cr(VI)) as potassium dichromate ($K_2Cr_2O_7$), three times per week for sixty days, starting on 1 July and finalizing on 30 August. The selection of the above Cr(VI) concentration was based on preliminary experiments using Cr(VI) at concentrations of 0–200 μM in order to determine the suitable Cr(VI) concentration for our experimentation and the corresponding treatment duration. During the vegetative period, the experimental vines were shaded using a green plastic net to avoid position effects on light intensity and to protect the vines from the harmful effects of high temperature and different meteorological hazards. For the water potential and photosynthetic parameter measurements, the green plastic net could be quickly removed so that maximum leaf exposure to light could be achieved when necessary. The rootstocks used in our experiment were chosen based on their importance in viticulture and the diversity of their genetic and agronomic characteristics. The 1103P rootstock was a Berlandieri–Rupestris hybrid, resistant to drought and relatively tolerant of iron chlorosis and salinity, whereas the 101-14 Mgt was a Riparia–Rupestris hybrid, sensitive to drought and to iron chlorosis, as well as to salinity.

The leaf chlorophyll content, stem water potential, and photosynthetic parameters were measured after the beginning of Cr(VI) application in three different stages: (1 July, 1 August, and 1 September, respectively).

### 2.2. Chlorophyll Content

The chlorophyll content documentation procedure in different experimental plots was realized using a CCM-200 chlorophyll content meter (CM) (Opti-Sciences, Tyngsboro, MA, USA). The CCM-200 is capable to provide instant and nondestructive onsite measurements of chlorophyll. Three fully expanded leaves from each plot and sampling time located on the basal nodes of the shoots were selected and marked. After measuring with the CCM-200, the marked leaves were cut, and the leaf chlorophyll contents were determined using a spectrophotometer (Jenway Ltd., Essex, UK) in typical laboratory conditions according to Wintermans and De Mots [19].

For the chlorophyll extraction, 0.5 g of fresh leaf-blade material was placed in 15 mL of ethanol (96%) and, subsequently, in a water bath at a temperature of 79.8 °C until complete discoloration occurred after approximately 2 h. The leaf-blade material was cut from the same leaf areas where the CCM-200 readings were performed. The chlorophyll a and b concentrations of the aliquot were measured at 665 and 649 nm, respectively, and the results were expressed on a fresh-weight (fw) basis according to the following equation:

$$Cl\ (a + b)\ mg\ g^{-1}\ FW = (6.10 \times A665 + 20.04 \times A649) \times 15/100\ FW \tag{1}$$

### 2.3. Stem Water Potential and Gas Exchange

Stem water potential (Ws) was measured at three stages during the period of the experiment on the same days and at the same time using a pressure chamber according to Chone et al. (2001) [20]. The Ws was measured for each vine on one specific, mature leaf that had been previously wrapped in a plastic bag to eliminate transpiration. Additionally, aluminum foil was placed around the plastic bag for at least 90 min prior to the measurements.

The PSII chlorophyll fluorescence parameters were recorded with the help of a chlorophyll fluorescence meter in attached, 30 min dark-adapted leaves with a portable chlorophyll fluorometer (PEA Hansatech Instruments Ltd., King's Lynn, UK) between 11:00 a.m. to 01:00 p.m. Artificial light exposure was realized using an actinic light source (635 nm) with a solid weak pulse of 3500 μ mols $m^{-2}\ s^{-1}$ PPFD (photosynthetic photon flux density). The light emitted from the LED source was filtered using an NIR filter to block any infrared spectrum components that could be received by the detector. The minimum ($F_0$), maximum (Fm), and variable (Fv) fluorescence yield parameters were automatically measured, and the maximum quantum yield (Fv/Fm) was recorded.

The net $CO_2$ assimilation rate (A) and stomatal conductance (gs) measurements were made on intact, well-developed leaves adjacent to those used for Ws between 11–13 h with a LCi portable gas exchange system (ADC BioScientific Ltd., Hoddesdon, UK).

### 2.4. Tissue Nutrient Concentrations and Growth Parameters

During harvest (early September), the experimental plants were uprooted and dissected into leaves, stems, trunks, and roots. The roots were separated from the potting medium by submerging the root balls in water and teasing the sand away from the roots. The dry weights (g) of the different parts of the vines were recorded. To determine the mineral composition, the samples were washed, dried at 70 °C for 48 h, and then ground to a fine powder to pass through a 30-mesh screen. Each sample (0.5 g) was dry-ashed at 520 °C for 5 h, and each one was dissolved in 3 mL of 6 N HCl and diluted with double-distilled water up to 50 mL. The concentrations of P, K, Ca, Mg, Na, Fe, Mn, Zn, and Cr were determined with ICP-OES (Perkin Elmer-Optical Emission Spectrometer, OPTIMA 2100 DV, Ontario, ON, Canada) [21], while those of N and B were determined by the Kjeldahl [22] and azomethine-H [23] methods, respectively.

### 2.5. Total Phenolics

During harvest, sixty days after the beginning of the Cr(VI) treatment, leaf sables were collected so that a total phenolic determination could be performed. The phenolics were extracted from 0.3 g of fresh leaf tissue submerged in 80% methanol and were determined according to the Folin–Ciocalteu colorimetric method [24]. Catechin was used to develop standard curves. The phenolic concentrations were expressed in mg·g$^{-1}$ catechin equivalents (CEs) of leaf fresh weight (f.w.).

### 2.6. Statistical Analysis

An analysis of variance was performed using SPSS Version 25 software, and the mean values were separated with the least significant difference.

## 3. Results

### 3.1. Plant Growth Parameters

Grapevines after exposure to 120 µM Cr(VI) for sixty days exhibited toxicity symptoms in terms of stunted root and shoot growth, leaf chlorosis (Figure 1), and discoloration of the roots (Figure 2).

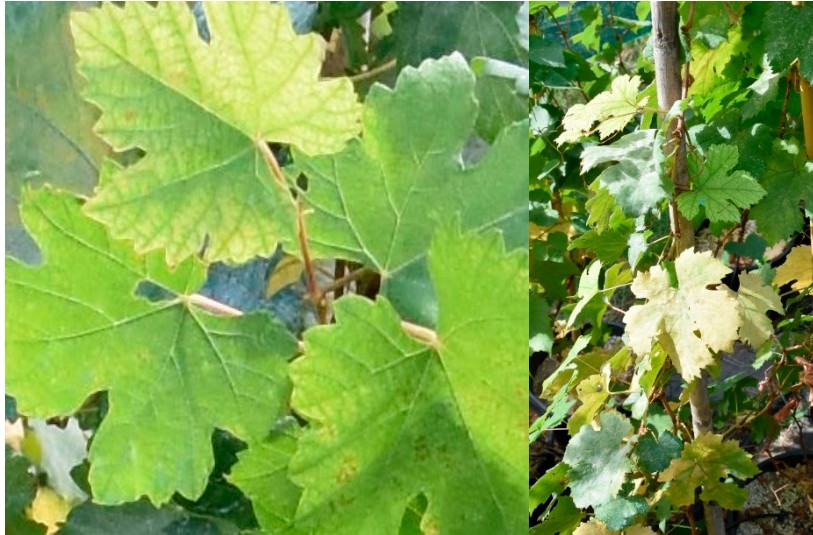

**Figure 1.** Cr(VI)-induced leaf chlorosis (left: first appearance in young leaves 45 days after the beginning of Cr(VI) treatment; right: overall leaf chlorosis 60 days after the beginning of Cr(VI) treatment).

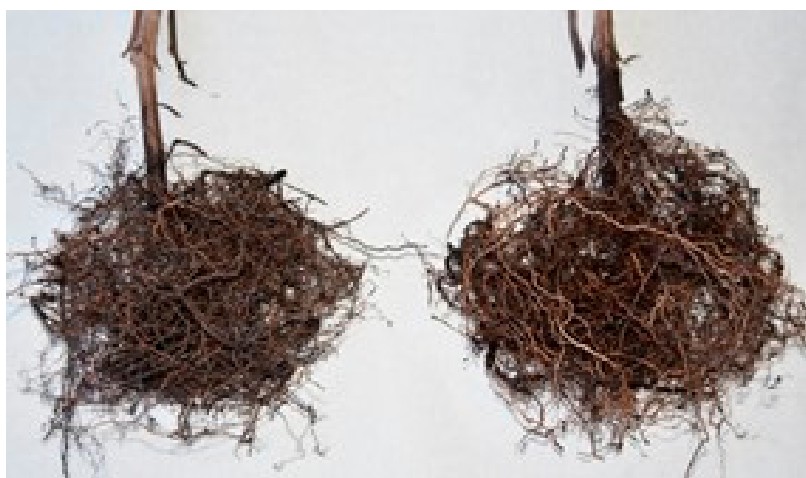

**Figure 2.** Root browning sixty days after the beginning of Cr(VI) treatment (left: Cr(VI) treatment; right: control).

The analyzed data showed a significant reduction in root and shoot dry weight (Table 1).

**Table 1.** The effect of Cr(VI) stress on shoot, trunk, and root dry weights of own-rooted Merlot and Cabernet Franc vine cultivars and vines grafted to 1103P and 101-14 Mgt rootstocks.

| | | Merlot | | | Cabernet Franc | | |
|---|---|---|---|---|---|---|---|
| | [g] | Shoots | Trunks | Roots | Shoots | Trunks | Roots |
| Control | Own roots | 49.78 [a] | 62.33 [a] | 52.83 [b] | 42.34 [ab] | 54.61 [c] | 44.54 [b] |
| | 1103P | 50.66 [a] | 59.09 [a] | 62.43 [a] | 43.12 [a] | 71.53 [a] | 51.32 [a] |
| | 101-14 Mgt | 43.63 [bc] | 45.11 [b] | 51.16 [b] | 38.78 [bc] | 62.35 [b] | 42.82 [b] |
| 120 µM Cr(VI) | Own roots | 46.16 [b] | 61.61 [a] | 44.87 [c] | 39.56 [ab] | 55.16 [c] | 37.16 [c] |
| | 1103P | 43.13 [bc] | 58.86 [a] | 32.37 [d] | 38.67 [bc] | 72.36 [a] | 41.34 [b] |
| | 101-14 Mgt | 41.66 [c] | 46.50 [b] | 31.67 [d] | 35.33 [c] | 60.94 [bc] | 31.53 [d] |
| | F: | 5.281 | 11.665 | 15.439 | 5.281 | 11.665 | 15.439 |

Different letters in each column represent significant difference at the $p < 0.05$.

The analysis of variance revealed a significant effect of Cr(VI) toxicity on growth parameters concerning mainly the root and shoot dry weights, whereas no significant effects were found in the trunk dry weights (Table 1). The differences between rootstocks and scion cultivars, as well as the rootstock–scion cultivar interaction, were found significant ($p < 0.001$) in terms of the shoot weights. Increased plant vigor was shown for the Merlot cultivar and for vines grafted to 1103P rootstocks. No significant rootstock–scion cultivar interaction was found for the root dry weight. Control vines grafted onto 1103P presented increased root weights when compared to the 101-14 Mgt rootstocks. On the contrary, no significant differences were observed in the Cr-treated vines. The results showed that Cr(VI) treatment significantly decreased the root and shoot dry weights. However, the reduction was more pronounced in the roots and less in the shoots. Decreased root dry weights resulted in increased shoot/root ratios in Cr(VI)-treated vines. According to the results, the shoot/root ratios ranged from 0.81 to 0.94 in the control vines and from 1.03 to 1.33 in the Cr-treated ones.

With regard to the visual symptoms of Cr toxicity, it was also observed that vines treated with 120 µM Cr(VI) manifested toxicity symptoms of interveinal chlorosis in young and middle leaves at approximately 45 days from the beginning of Cr(VI) treatment. On the 60th day, at the end of experimentation, a general leaf chlorosis was observed in all the vines. In addition, as shown in Figure 2, Cr(VI) toxicity caused a brown coloration of the roots.

### 3.2. Tissue Nutrient Concentrations

In the case of the macronutrients, Cr(VI) stress caused a significant reduction in all the nutrient concentrations. The degree of reduction varied according to scion and rootstock genotype (Table 2).

**Table 2.** The effect of Cr(VI) toxicity on macronutrient (N, P, K, Ca, and Mg) concentrations in different plant compartments of Merlot and Cabernet Franc vines on own roots or grafted to 1103P and 101-14 Mgt rootstocks.

| | | Leaves | | | | | | | | | |
|---|---|---|---|---|---|---|---|---|---|---|---|
| | | Merlot | | | | | Cabernet Franc | | | | |
| | [% d.w.] | N | P | K | Ca | Mg | N | P | K | Ca | Mg |
| Control | Own roots | 2.13 a | 0.31 a | 2.12 a | 3.07 a | 0.57 a | 2.56 b | 0.30 abc | 2.31 a | 3.16 a | 0.55 a |
| | 1103P | 2.09 a | 0.30 a | 1.47 cd | 2.58 b | 0.46 ab | 2.95 a | 0.35 a | 1.46 b | 2.98 ab / 2.98 ab | 0.47 abc |
| | 101-14 Mgt | 1.79 b | 0.29 a | 1.83 b | 2.62 b | 0.52 ab | 1.81 cd | 0.32 ab | 2.03 a | 2.89 bc | 0.54 ab |
| 120 μM Cr(VI) | Own roots | 1.79 b 1. | 0.21 b | 0.75 e | 2.74 b | 0.39 bc | 1.58 d | 0.29 bc | 1.30 b | 2.73 cd | 0.41 bc |
| | 1103P | 1.80 b | 0.25 ab | 1.40 d | 2.36 c | 0.41 bc | 1.92 c | 0.27 bc | 1.21 b | 2.62 d | 0.38 c |
| | 101-14 Mgt | 1.81 ab | 0.20 b | 1.59 c | 2.62 b | 0.32 c | 1.73 cd | 0.26 c | 1.42 b | 2.67 d | 0.41 bc |
| | F | 8.344 | 6.502 | 44.952 | 9.699 | 6.149 | 8.344 | 6.502 | 44.952 | 9.699 | 6.194 |
| | | Shoots | | | | | | | | | |
| | | Merlot | | | | | Cabernet Franc | | | | |
| | | N | P | K | Ca | Mg | N | P | K | Ca | Mg |
| Control | Own roots | 0.32 e | 0.20 bc | 0.65 a | 0.78 a | 0.17 b | 0.87 a | 0.24 ab | 0.61 a | 0.46 c | 0.16 b |
| | 1103P | 0.71 bc | 0.24 ab | 0.63 ab | 0.79 a | 0.15 c | 0.85 ab | 0.29 a | 0.67 a | 0.77 a | 0.17 b |
| | 101-14 Mgt | 0.76 b | 0.30 a | 0.58 ab | 0.75 ab | 0.65 a | 0.84 abc | 0.23 abc | 0.63 a | 0.73 a | 0.26 a |
| 120 μM Cr(VI) | Own roots | 0.87 a | 0.18 bc | 0.38 ab | 0.68 bc | 0.14 c | 0.77 c | 0.16 cd | 0.42 ab | 0.57 b | 0.13 c |
| | 1103P | 0.68 cd | 0.19 bc | 0.36 b | 0.69 bc | 0.15 c | 0.79 bc | 0.21 bc | 0.25 b | 0.59 b | 0.14 c |
| | 101-14 Mgt | 0.62 d | 0.16 c | 0.43 ab | 0.65 c | 0.14 c | 0.64 d | 0.12 d | 0.29 b | 0.53 bc | 0.13 c |
| | F | 7.571 | 1.466 | 15.131 | 1.860 | 44.395 | 7.571 | 1.466 | 15.131 | 1.860 | 44.395 |
| | | Trunks | | | | | | | | | |
| | | Merlot | | | | | Cabernet Franc | | | | |
| | | N | P | K | Ca | Mg | N | P | K | Ca | Mg |
| Control | Own roots | 0.61 c | 0.28 a | 0.35 b | 0.90 a | 0.15 b | 0.63 b | 0.26 a | 0.24 bc | 0.94 a | 0.18 a |
| | 1103P | 0.71 a | 0.19 b | 0.27 c | 0.89 a | 0.16 ab | 0.64 b | 0.20 b | 0.25 ab | 0.91 ab | 0.16 b |
| | 101-14 Mgt | 0.687 ab | 0.17 b | 0.43 a | 0.83 a | 0.17 a | 0.71 a | 0.23 ab | 0.28 a | 0.82 bc | 0.14 c |
| 120 μM Cr(VI) | Own roots | 0.51 d | 0.08 c | 0.28 c | 0.62 b | 0.12 c | 0.55 c | 0.16 c | 0.23 bc | 0.74 cd | 0.12 d |
| | 1103P | 0.56 cd | 0.11 c | 0.25 cd | 0.81 a | 0.13 c | 0.65 ab | 0.13 cd | 0.19 d | 0.76 cd | 0.13 cd |
| | 101-14 Mgt | 0.63 bc | 0.09 c | 0.22 d | 0.56 b | 0.15 b | 0.60 bc | 0.11 d | 0.21 cd | 0.68 d | 0.13 cd |
| | F | 2.550 | 16.739 | 5.185 | 4.937 | 6.972 | 2.550 | 16.739 | 5.186 | 4.937 | 6.972 |
| | | Roots | | | | | | | | | |
| | | Merlot | | | | | Cabernet Franc | | | | |
| | | N | P | K | Ca | Mg | N | P | K | Ca | Mg |
| Control | Own roots | 1.05 | 0.26 b | 0.60 a | 1.18 | 0.24 b | 1.12 | 0.30 a | 0.65 a | 0.94 | 0.27 a |
| | 1103P | 0.99 | 0.28 a | 0.50 a | 1.39 | 0.29 a | 1.06 | 0.28 b | 0.48 b | 0.90 | 0.25 ab |
| | 101-14 Mgt | 1.05 | 0.23 c | 0.53 a | 1.03 | 0.24 b | 1.10 | 0.27 b | 0.42 bc | 1.05 | 0.26 a |
| 120 μM Cr(VI) | Own roots | 0.83 | 0.22 c | 0.38 b | 1.24 | 0.22 bc | 0.98 | 0.22 cd | 0.39 bc | 1.16 | 0.27 a |
| | 1103P | 0.85 | 0.18 d | 0.27 c | 0.96 | 0.20 c | 0.87 | 0.21 d | 0.24 d | 1.24 | 0.25 ab |
| | 101-14 Mgt | 0.88 | 0.22 c | 0.50 a | 0.96 | 0.19 c | 0.89 | 0.23 c | 0.36 c | 1.05 | 0.22 b |
| | F | | 12.907 | 8.597 | | 2.663 | | 12.907 | 8.597 | | 2.663 |

Different letters in each column represent significant difference at the $p < 0.05$.

Nitrogen (N) concentrations ranged from 1.79 to 2.56% in the leaves of the control vines and from 1.58 to 1.92% in Cr(VI)-treated vines. Moreover, the N concentration was decreased by 32.48% in Cr(VI)-treated vines compared to the control vines. Lower N concentrations were also detected in the other parts of the vines. However, no significant changes were observed in the roots. Phosphorus concentrations ranged from 0.20 to 0.35% in the leaves, from 0.12 to 0.30% in the shoots, from 0.08 to 0.28% in the trunks, and from 0.18 to 0.30% in the roots. The highest value (0.35%) was found in the leaves of

control Merlot vines grafted onto 1103P rootstocks and the lowest value (0.08%) in the trunks of own-rooted, treated Merlot vines. Vines exposed to Cr(VI) toxicity presented significantly decreased P concentrations in all the cases. Under chromium(VI) treatment, the K concentrations decreased in all parts of the vines. The highest and lowest K concentration values were recorded in the grapevine leaves and trunks, respectively. Higher K values in the leaves were recorded in own-rooted vines compared to the grafted vines. Regarding the rootstock effect, it was found that the vines raised on 101-14 Mgt rootstocks under chromium(VI) toxicity preserved higher K levels of 15.38% in their leaves compared to those raised on 1103P rootstocks. In addition, according to our results, Cabernet Franc accumulated higher K concentrations in the leaves than Merlot.

The highest Ca concentration was found in the leaves (spanning from 2.62 to 3.16%). The effects of rootstock and scion variety were important. Interestingly, it was reported that the 101-14 Mgt rootstock with Cabernet Franc accumulated higher amounts of Ca in the leaves. Lower concentrations were found in the other parts of the vines. The exposure of vines to Cr(VI) stress caused a significant decrease in the leaf, shoot, and trunk Ca concentrations. However, the rate of decrease was drastically higher in trunks (26%) than in other parts. No significant differences were found in the roots between the treated and control samples.

Magnesium concentrations ranged from 0.38 to 0.55% in the leaves, 0.13 to 0.26% in the shoots, 0.12 to 0.18% in the trunks, and 0.22 to 0.27% in the roots. Foliar Mg was significantly affected by the rootstock; vines grafted onto 101-14 Mgt rootstocks showed higher Mg concentrations. Cr(VI) stress reduced Mg concentrations in all parts of the vines (from approximately 31% in the shoots to 7% in the roots).

As shown in Table 3, chromium(VI) treatment significantly increased the total chromium (Cr) concentrations in the discrete sections of the vines.

It was found that the root Cr concentrations were significantly higher than those in all the above-ground vine parts. In the case of the above-ground parts of the vines that were treated with Cr, the lowest Cr concentration was recorded in the shoots. The recorded Cr concentrations in the leaves and trunks were accordingly smaller. The shoots and leaves of chromium(VI)-treated plants contained 296 and 209 times less Cr than the roots, respectively. Regarding rootstock effect, a significantly increased Cr concentration was recorded in Cabernet Franc roots when grafted onto 101-14 Mgt rootstocks versus using the 1103P rootstocks. The Cr transfer from vine roots to aerial tissues was limited, with decreasing concentrations in the shoots, leaves, trunks, and roots, respectively. Additionally, grafted vines accumulated less Cr in their roots and shoot tissues when compared to the nongrafted ones.

The results collected that related to B, Fe, Zn, Mn, and Cu micronutrient concentrations in the leaves, roots, shoots, and trunks are shown in Table 3. Among the different parts of the vines, the roots showed higher Fe and Zn concentrations, whereas leaves concentrated more B. The mean boron concentrations ranged from 50.59 in the leaves to 9.93 mg kg$^{-1}$ in the trunks.

A significant decrease in the uptake of micronutrients such as boron (B), iron (Fe), zink (Zn), manganese (Mn), and copper (Cu) was found under excessive Cr(VI) exposure (Table 3). According to the results, Cr toxicity decreased iron concentrations from 17.3% to 35.6% and B concentrations from 13.80% to 27.21% in different grapevine parts.

**Table 3.** The effect of Cr(VI) toxicity on micronutrient and metal (Fe, Mn, Zn, B, Cu, and Cr) concentrations in different parts of Merlot and Cabernet Franc vines on own roots or grafted to 1103P and 101-14 Mgt rootstocks.

| | | Leaves | | | | | | | | | | | |
| | | Merlot | | | | | | Cabernet Franc | | | | | |
| | [mg kg$^{-1}$] | Fe | Mn | Zn | B | Cu | Cr | Fe | Mn | Zn | B | Cu | Cr |
|---|---|---|---|---|---|---|---|---|---|---|---|---|---|
| Control | Own roots | 107.89 ab | 68.23 b | 18.26 a | 56.84 ab | 9.89 a | 0.47 c | 112.37 bc | 62.69 a | 13.42 b | 46.03 ab | 8.35 a | 0.63 c |
| | 1103P | 121.29 a | 96.04 a | 12.86 c | 62.17 a | 7.71 b | 0.55 c | 136.27 a | 63.88 a | 12.73 bc | 51.92 a | 6.02 bc | 0.16 c |
| | 101-14 Mgt | 100.56 b | 52.90 c | 13.00 c | 51.06 b | 6.12 bc | 0.69 c | 122.76 ab | 48.20 bc | 15.50 a | 45.91 ab | 6.45 ab | 0.62 c |
| 120 µM Cr(VI) | Own roots | 96.81 b | 48.42 cd | 14.05 b | 46.37 b | 5.02 c | 3.49 a | 102.53 cd | 56.55 ab | 11.80 c | 41.84 b | 5.34 bc | 3.01 ab |
| | 1103P | 94.74 b | 39.26 de | 11.47 d | 44.85 b | 4.11 c | 2.01 b | 92.03 d | 48.37 bc | 8.75 d | 45.33 ab | 4.21 c | 2.15 b |
| | 101-14 Mgt | 105.47 b | 36.69 e | 11.85 d | 47.35 b | 4.81 c | 3.22 a | 90.89 d | 39.97 c | 9.21 d | 40.37 b | 4.32 c | 3.28 a |
| | F | 10.024 | 23.921 | 25.634 | 24.486 | 11.889 | 28.305 | 10.024 | 23.921 | 25.634 | 24.486 | 11.889 | 28.305 |
| | | Shoots | | | | | | | | | | | |
| | | Merlot | | | | | | Cabernet Franc | | | | | |
| | | Fe | Mn | Zn | B | Cu | Cr | Fe | Mn | Zn | B | Cu | Cr |
| Control | Own roots | 24.27 a | 44.84 ab | 30.79 a | 15.78 a | 10.46 a | 0.66 b | 25.90 a | 30.42 b | 24.90 b | 13.43 b | 7.62 bc | 0.75 b |
| | 1103P | 21.13 a | 43.27 b | 22.42 b | 15.00 a | 7.99 a | 0.54 b | 23.77 ab | 46.70 a | 31.67 a | 14.24 b | 10.71 a | 0.59 b |
| | 101-14 Mgt | 24.69 a | 50.03 a | 31.35 a | 14.96 a | 10.27 a | 0.48 b | 24.08 ab | 40.83 a | 16.25 cd | 16.55 a | 8.31 ab | 0.52 b |
| 120 µM Cr(VI) | Own roots | 15.92 b | 24.74 d | 14.79 d | 11.03 c | 7.54 b | 1.90 ab | 21.70 b | 29.71 b | 14.86 d | 9.39 c | 5.30 c | 2.98 a |
| | 1103P | 15.07 b | 34.83 c | 14.02 d | 13.18 b | 6.64 b | 2.47 a | 15.47 c | 28.55 b | 17.36 c | 9.10 c | 5.13 c | 2.06 ab |
| | 101-14 Mgt | 17.09 b | 44.22 b | 20.33 c | 12.85 b | 7.63 a | 2.60 a | 16.95 c | 21.72 c | 10.29 e | 9.77 c | 5.64 bc | 2.11 ab |
| | F | 6.661 | 15.88 | 33.063 | 15.281 | 27.921 | 37.616 | 6.661 | 15.88 | 33.063 | 15.281 | 27.921 | 37.616 |
| | | Trunks | | | | | | | | | | | |
| | | Merlot | | | | | | Cabernet Franc | | | | | |
| | | Fe | Mn | Zn | B | Cu | Cr | Fe | Mn | Zn | B | Cu | Cr |
| Control | Own roots | 110.03 a | 24.40 b | 18.95 ab | 11.55 a | 8.29 abc | 3.18 b | 101.49 a | 18.57 b | 21.10 a | 11.48 a | 8.96 abc | 4.50 c |
| | 1103P | 107.37 ab | 27.02 a | 15.72 abc | 10.53 b | 9.92 a | 4.19 b | 109.38 a | 24.97 a | 18.92 ab | 11.01 ab | 9.82 a | 3.34 c |
| | 101-14 Mgt | 99.24 ab | 25.85 ab | 20.07 a | 10.46 b | 9.49 ab | 5.24 b | 97.67 ab | 23.81 a | 19.30 ab | 10.58 a | 9.54 ab | 3.64 c |
| 120 µM Cr(VI) | Own roots | 58.75 c | 18.19 d | 10.26 bc | 9.14 c | 5.47 c | 95.28 a | 73.13 abc | 19.21 b | 9.52 c | 8.41 c | 5.98 bc | 102.24 a |
| | 1103P | 78.40 abc | 23.63 b | 10.58 bc | 8.96 c | 6.01 bc | 58.76 ab | 54.27 c | 19.61 b | 10.52 bc | 8.92 c | 5.55 c | 40.95 bc |
| | 101-14 Mgt | 70.11 bc | 20.75 c | 9.89 c | 9.01 c | 5.89 bc | 81.28 a | 60.80 bc | 18.57 b | 9.24 c | 8.96 c | 5.46 c | 62.23 ab |
| | F | 14.320 | 6.983 | 25.058 | 7.903 | 10.334 | 16.902 | 14.32 | 6.983 | 25.058 | 7.903 | 10.334 | 16.902 |
| | | Roots | | | | | | | | | | | |
| | | Merlot | | | | | | Cabernet Franc | | | | | |
| | | Fe | Mn | Zn | B | Cu | Cr | Fe | Mn | Zn | B | Cu | Cr |
| Control | Own roots | 213.24 ab | 32.22 a | 26.72 b | 20.88 c | 26.23 a | 11.3 c | 230.50 a | 35.27 a | 22.34 c | 24.99 b | 31.76 ab | 11.94 d |
| | 1103P | 202.96 abc | 27.79 b | 30.61 a | 22.20 b | 25.84 a | 6.22 c | 204.52 ab | 26.19 b | 29.23 a | 29.82 b | 24.31 abc | 9.71 d |
| | 101-14 Mgt | 245.38 a | 35.90 a | 30.37 a | 23.24 a | 30.90 a | 11.63 c | 231.23 a | 32.41 ab | 24.81 b | 23.70 c | 33.14 a | 11.85 d |
| 120 µM Cr(VI) | Own roots | 160.63 bcd | 22.83 bc | 15.69 c | 19.34 d | 19.96 ab | 658.81 a | 163.76 b | 27.66 b | 15.17 d | 24.77 b | 17.89 c | 868.34 a |
| | 1103P | 136.24 d | 29.04 b | 16.97 c | 19.19 d | 12.95 b | 396.05 b | 177.60 ab | 30.01 ab | 16.81 d | 23.21 c | 18.56 c | 570.76 c |
| | 101-14 Mgt | 151.13 cd | 20.23 c | 17.15 c | 16.95 e | 18.70 ab | 427.19 b | 175.52 ab | 28.16 b | 17.19 d | 23.11 c | 19.64 bc | 718.45 b |
| | F | 5.262 | 2.535 | 16.015 | 5.532 | 11.928 | 117.731 | 5.262 | 2.535 | 16.015 | 5.532 | 11.928 | 117.73 |

Different letters in each column represent significant difference at the *p* < 0.05.

### 3.3. Chlorophyll Content

The foliar content of the total chlorophyll was assayed under Cr(VI) treatment and showed significantly decreased concentrations. The leaf chlorophyll content was determined on the 1st, 30th, and 60th days of the experimentation using both analytical and nondestructive methods (CCM-200 index). We constructed graphs based on the results (Figure 3).

Cr(VI) reduced both the leaf chlorophyll concentration and the CCM-200 chlorophyll index, mainly at 30 and 60 days from the beginning of the experimentation. The leaf total chlorophyll concentration in Cr-treated vines decreased from 45.23% on the 30th day to 53.46% on the 60th day. At these two stages, the chlorophyll reduction was significantly higher in vines grafted to 101-14 Mgt rootstocks when compared to the 1103P rootstocks. Sixty days after the beginning of Cr treatment, the chlorophyll concentrations were least in the case of the 101-14 Mgt rootstock, whereas higher values were recorded in 1103P rootstocks. Similar to chlorophyll concentration, the CCM-200 index was affected by chromium toxicity (Figure 3). A similar trend for the relative chlorophyll content was also observed.

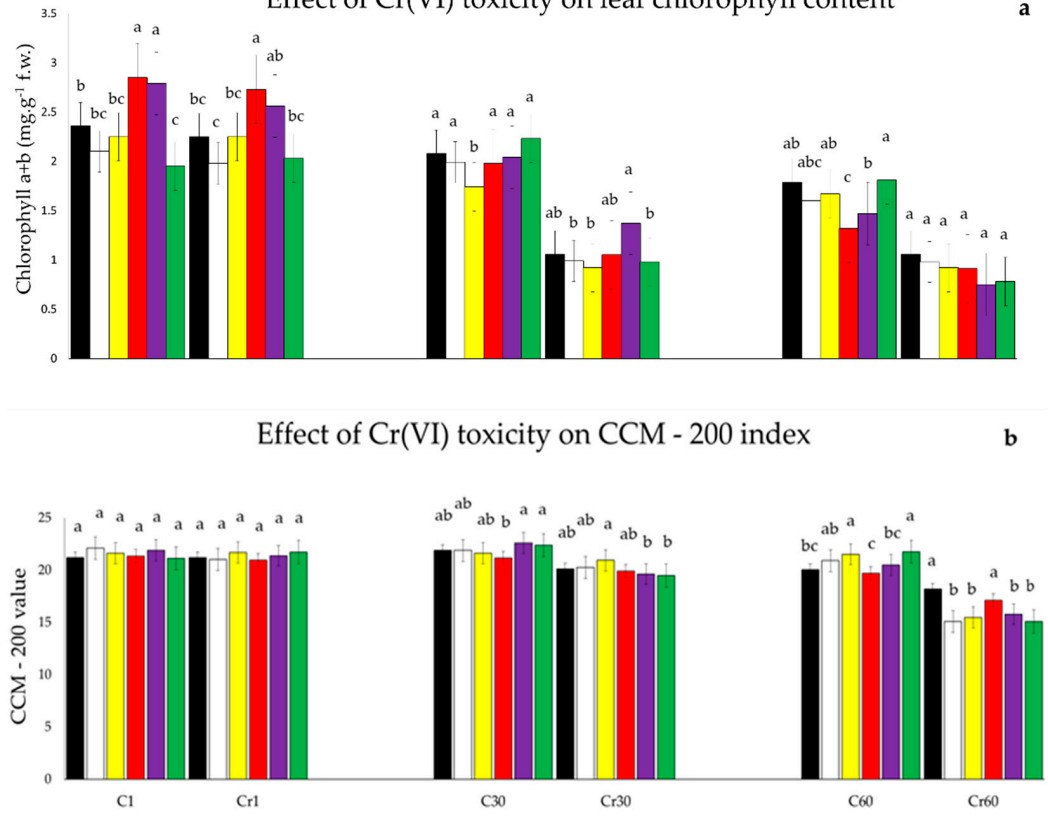

**Figure 3.** Leaf chlorophyll content (**a**) and CCM-200 chlorophyll index (**b**) at 1st, 30th, and 60th days after the beginning of Cr(VI) treatment in own-rooted (OR) Merlot (M) and Cabernet Franc (CF) vine varieties or vines grafted to 1103P and 101-14 Mgt rootstocks. C: control; Cr: Cr(VI) treatment; 1,30,60: at days one, thirty, and sixty respectively. Chl a+b (F:23.191). CCM-200 (F: 10.319). Different letters represent significant difference of $p < 0.05$.

### 3.4. Total Phenolic Content

The effect of Cr(VI) toxicity on the leaf total phenolic content is presented in Figure 4.

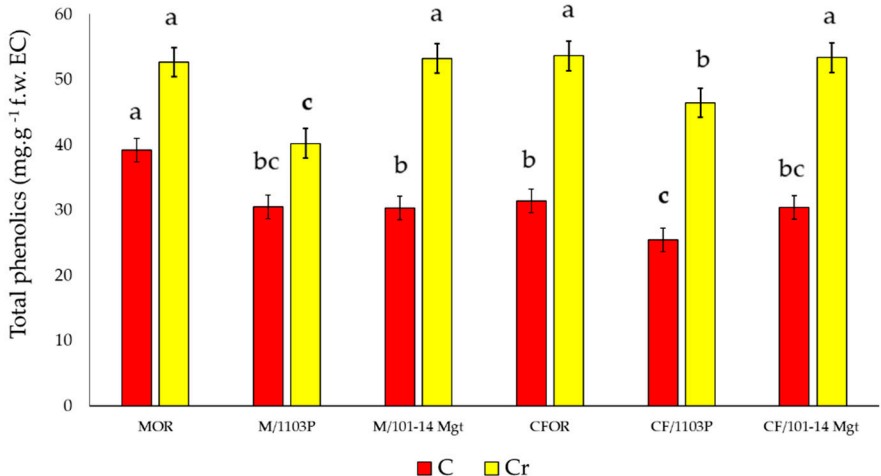

**Figure 4.** Leaf total phenolic content after sixty days of Cr(VI) treatment in own-rooted (OR) Merlot (M) and Cabernet Franc (CF) vine varieties and vines grafted to 1103P and 101-14 Mgt rootstocks. C: control, Cr: Cr(VI) treatment. (F:26.14). Different letters represent significant difference of $p < 0.05$.

In the examined samples, sixty days from the beginning of experimentation the leaf total phenolic contents ranged from 25.42 to 53.35 mg·g$^{-1}$. With Cr(VI) treatment, the accumulation of these compounds was significantly increased from 46.14% in Merlot vines to 75.91% in Cabernet Franc vines. According to the analysis of variance, the effects of Cr(VI) treatment and rootstock were significant, whereas no significant differences were found between the scion varieties.

### 3.5. Water Status and Photosynthetic Activity

The stem water potential (SWP) decreased gradually during the period of the experimentation, reaching minimum values after sixty days of Cr(VI) treatment (Figure 5).

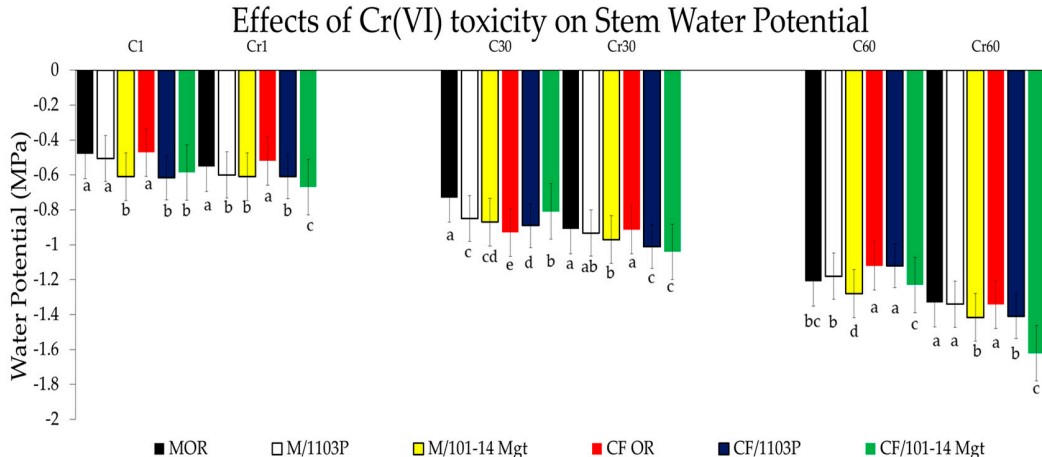

**Figure 5.** Chromium(VI) toxicity effects on midday stem water potential of Merlot (M) and Cabernet Franc (CF) grapevine cultivars on their own roots (OR) and on 1103P and 101-14 Mgt rootstocks at three stages. C: control; Cr: Cr(VI) treatment; 1, 30, 60 (at days one, thirty, and sixty, respectively). F: 85.711. Different letters represent significant difference of *p* < 0.05.

The stem water potential declined continuously during the Cr(VI) treatment period (Figure 5). The minimum values were recorded at the end of the Cr(VI) treatment cycle (60 days). Thirty and sixty days after the start of the Cr(VI) treatments, the stem water potential was significantly decreased in all cases of grafted or own-rooted vines. Moreover, the effects of the rootstock and Cr(VI) treatment were significant. Between the two rootstocks used in our experiment, 101-14 Mgt always rated lower values of stem water potential compared to 1103P. On the other hand, significant differences were found between the Cr-stressed and the control vines.

The photosynthetic activity was evaluated by measuring the net assimilation rate (A), stomatal conductance (gs), and PSII chlorophyll fluorescence (ChF). The responses of net photosynthesis rate (A) and stomatal conductance (gs) in the chromium-stressed vines are shown in the Figure 6.

The effects of Cr(VI) toxicity on the net $CO_2$ assimilation rate (A) and stomatal conductance (gs) are presented in Figure 6. The vines exposed to 120 μM Cr(VI) for thirty or sixty days showed decreased photosynthetic activity. No statistically different values from the control plants were found at the beginning of the experimental cycle (data not shown). According to the analysis of variance, the effects of rootstock, scion cultivar, and Cr(VI) treatment, as well as their interactions, were significant (*p* < 0.001). Generally, in Cr(VI)-stressed vines, the net $CO_2$ assimilation rate was reduced, ranging from 5.11 to 7.25 μmol $CO_2$ m$^{-2}$ s$^{-1}$ on day thirty of the treatment cycle and from 2.54 to 4.9 on the last day. The leaf stomatal conductance was also significantly decreased in Cr(VI)-stressed vines (Figure 6b). However, no significant differences were found between the rootstocks used in the experiment and between the scion varieties.

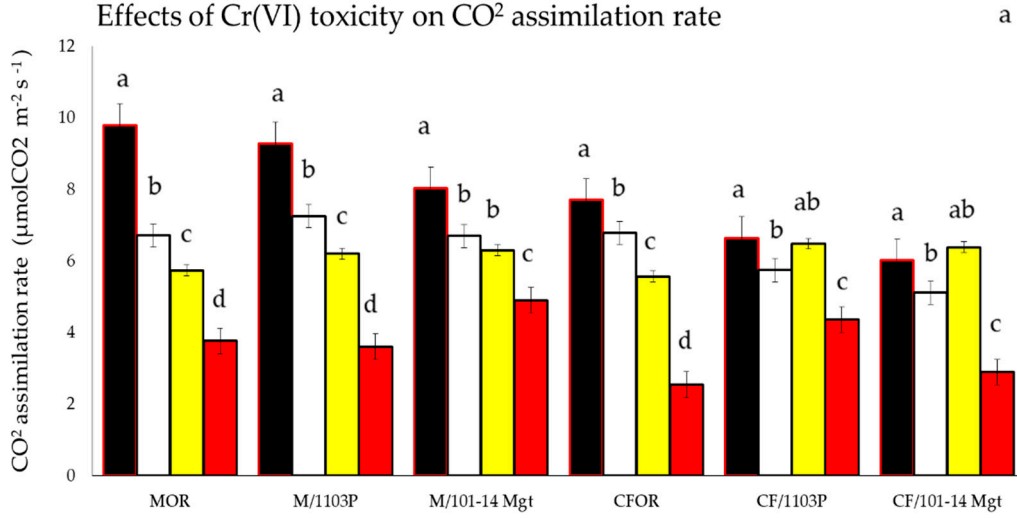

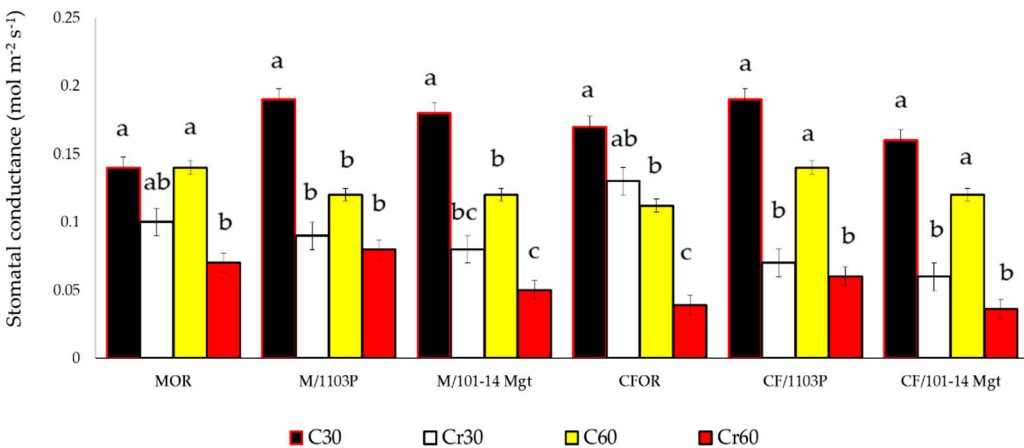

**Figure 6.** Net $CO_2$ assimilation rate (**a**) and stomatal conductance (**b**) of Merlot (M) and Cabernet Franc (CF) grapevine cultivars on their own roots (OR) and on 1103P and 101-14 Mgt rootstocks thirty and sixty days after the start of the Cr(VI) treatments. C: control; Cr: Cr(VI) treatment; 30 and 60 (at days thirty and sixty, respectively). Net assimilation rate (F:21.174). Stomatal conductance (F: 1.475). Different letters represent significant difference of $p < 0.05$.

The measurements of the chlorophyll fluorescence parameters characterizing the photochemical activity of the Cr-treated plants are summarized in Table 4. It was found that, during the first period of experimentation, the chlorophyll fluorescence parameters were not affected by the Cr(VI) treatment. A significant decrease in the Fv/Fm ratio was observed after 30 days of culturing when a small decrease in this parameter was observed. From day thirty to day sixty of the treatment cycle, the values of the Fv/Fm ratios were decreased, especially at day sixty. At this last stage, the Fv/Fm ratios were clearly strongly inhibited, showing the lowest values in the Cabernet Franc cultivars grafted to 101-14 Mgt rootstocks.

**Table 4.** Effects of Cr(VI) toxicity on maximum quantum yield of photosystem II (Fv/Fm) of Merlot (M) and Cabernet Franc (CF) grapevine cultivars on their own roots (OR) and on 1103P and 101-14 Mgt rootstocks.

|  |  | Merlot | | Cabernet Franc | |
|---|---|---|---|---|---|
|  |  | **Thirty Days** | **Sixty Days** | **Thirty Days** | **Sixty Days** |
| Control | Own roots | 0.783 [bc] | 0.721 [b] | 0.779 [b] | 0.721 [a] |
|  | 1103P | 0.820 [ab] | 0.732 [ab] | 0.791 [ab] | 0.744 [a] |
|  | 101-14 Mgt | 0.837 [a] | 0.771 [a] | 0.821 [a] | 0.716 [a] |
| 120 μM Cr(VI) | Own roots | 0.761 [c] | 0.675 [c] | 0.718 [d] | 0.576 [b] |
|  | 1103P | 0.800 [abc] | 0.501 [d] | 0.771 [bc] | 0.579 [b] |
|  | 101-14 Mgt | 0.773 [c] | 0.494 [d] | 0.736 [cd] | 0.482 [c] |
|  |  | F: 19.015 | | | |

Different letters in each column represent significant difference at the $p < 0.05$.

## 4. Discussion

As with other heavy metals, chromium(VI) may affect the most important physiological processes linked to plant growth and development, such as water relations, leaf gas exchange, nutrient uptake, and transport. The toxic effects of Cr(VI) are also evident, showing symptoms such as damaged roots, the reduction of root and shoot growth, leaf chlorosis, and the brown coloration of roots.

### 4.1. Nutrient Concentrations in Plant Tissues and Vine Growth

Root Cr values were considerably higher than those determined in the above-ground parts of vines (Table 3). This could be attributed to the immobilization of Cr in the vacuoles of root cells, rendering it less toxic, which may be a natural toxicity response of the plant [25]. According to Mangabeira et al. 2011 [26], the higher accumulation of Cr in roots might be attributed to the sequestration of Cr in the vacuoles of root cells as a protective mechanism. The cell walls contain proteins and polysaccharides (cellulose, hemicellulose, and pectin) with various chemical functional groups to effectively bind metal ions and act as the first barrier to limit heavy metal entrance into the cell [27]. Thus, this mechanism provides some natural tolerance in plants towards Cr toxicity. High Cr accumulation in root has been considered an exclusion response to limit Cr transport to aerial parts [28,29]. It was also reported that the reduced translocation of Cr to aerial plant parts may be due to the conversion of Cr(VI) to Cr(III) [30]. On the other hand, grafted vines accumulated lower amounts of total Cr in their roots and shoot tissues in comparison to nongrafted vines.

It is interesting to note that the Cr(VI)-treated vines accumulated restricted amounts of total chromium in the leaves and shoots. The leaf Cr concentrations in treated vines ranged between 2.49 and 3.28 $\mu g\,g^{-1}$ d.w., which was below the upper limits for human consumption. Likewise, according to our personal unpublished data, berries showed very low Cr concentrations (0.407–1.448 $\mu g\,g^{-1}$ d.w.). Therefore, these leaves and berries were both considered adequate for human consumption.

There have been a lot of reports on the effect of Cr toxicity on mineral uptake and Cr compartmentalization in plants. For instance, Cr concentration was observed to be highest in the cytoplasm and intercellular spaces of the root cell wall of *Iris pseudacorus* [28]. Furthermore, the translocation of Cr from the roots to the shoots is very limited and it depends on the chemical form of Cr inside the tissue [16]. In our experiment, vines exposed to 120 μM Cr(VI) only transferred 0.45% of the total Cr accumulated in the roots to the leaves. As regards the rootstock effect, we can see in Table 3 that own-rooted vines and those raised on 101-14 Mgt rootstocks showed higher Cr concentrations. Among the assessed rootstocks, it seems that 1103P was best excluder of Cr concentration in the roots and in other aerial parts of the vines.

Chromium accumulation in the roots and trunks, as in the other aerial parts, led to various physiological and biochemical changes, especially in nutrient uptake and transport.

In the Cr-stressed vines, both macronutrient and micronutrient uptake decreased. According to our results, N decreased by 32.48% in Cr-stressed vines compared to the control (Table 1). It was reported that, in most cases, heavy metals blocked the entry of N in the root system [31].

Cr(VI) treatment also markedly inhibited the uptake of P, K, Ca, Mg, B, Fe, Mn, Zn, and Cu. The reduction in nutrient uptake could be attributed to reduced root growth due to Cr toxicity (Tables 1 and 2). Alternatively, Cr might replace some nutrients at their physiological binding sites due to ionic resemblance, interfering with their uptake and translocation [32–34]. In addition, Cr can compete with other nutrients, such as Ca, Mg, Fe K, P, and N, leading to their reduced adsorption and uptake [35]. The observed decrease in Mg content was previously demonstrated [36]. The significance of the Mg element has been particularly established regarding the role of Mg in photosynthesis. The most familiar role of Mg in photosynthesis is as the central atom of the chlorophyll molecule. The insertion of Mg into the porphyrin structure during chlorophyll formation is catalyzed by Mg-chelatase [37]. It was reported that Cr can replace Mg ions at the active sites of many enzymes, such as δ-aminolevulinate acid dehydratase (ALAD), that are involved in chlorophyll biosynthesis [38]. It is well known that ALAD is a metalloenzyme, and its activity in plants is dependent on the availability of Mg [39]. We also recorded a significant decrease in the uptake of Fe in grapevine plants exposed to Cr(VI). The reduction in Fe content under Cr(VI) stress has been reported in several works on other plant species [40,41]. An earlier report suggested Cr interference in the availability of Fe leading to the impairment of Fe metabolism [42]. Furthermore, the indirect effect of Cr on chlorophyll reduction resulted in a depletion of Fe. Indeed, in Fe-deficient plants, destruction of the lamellar system of thylakoids and changes in the lipid and protein compositions of these membranes were reported [43].

Our results concerning nutrient uptake are in accordance with other ones [40,44]. Potassium is an essential nutrient for protein synthesis, glycolytic enzymes, and photosynthesis [45]. In our experiment, Cr(VI)-stressed vines raised on 101-14 Mgt rootstocks and Cabernet Franc scion cultivars accumulated increased K ions in their leaves compared to 1103P and Merlot vines.

The decrease in K could also result from an efflux of these nutrients due to membrane damage caused by excess lipid peroxidation, leading to increased permeability and reduced selectivity of the membranes [46]. On the other hand, the reduction in N, K, P, and other elements could be due to the reduced root growth induced by Cr(VI) toxicity. It was reported that nitrogen fertilization depressed the uptake and leaf concentration of chromium, similar to other metals [15].

The results of the present study showed that Cr-stressed vines had decreased shoot and root dry weights with a more pronounced reduction in root dry weight (Table 1). Roots, being the direct targets of the Cr exposure, showed many more toxicity symptoms than shoots. Poor root development leads to lower nutrient availability for shoots; thus, the shoots also show poor development [3,47]. According to the following discussion, Cr(VI) toxicity induced negative effects on the photosynthetic parameters, but the highest reduction was observed after 60 days of Cr(VI) treatment. On the other hand, the decreased root dry weights resulted to increased shoot/root ratios in Cr(VI)-treated vines. In addition to stunted root growth, a brown coloration was observed as a visual symptom of Cr toxicity in the root system (Figure 2) at the end of the experiment (60 days of Cr stress). Cr(VI) was reported to cause brown coloration of the roots, stunted root growth, impaired root hair formation, and enhanced lateral root growth [48]. Decreased root growth in response to Cr(VI) was proposed due to reduced cell division or cell elongation in the root tips [44]. Other studies have also reported a decreased mitotic index in growing root tips due to Cr toxicity [49]. In our experiment, visual toxicity symptoms were observed in the leaves at about 45 days from the beginning of Cr(VI) treatment (Figure 1). Interveinal chlorosis first appeared in the young and middle leaves of own-rooted vines, followed by grafted ones. At the end of the experimentation, a general leaf chlorosis appeared in all the vines.

Leaf chlorosis and necrosis has been reported for other plant species in several works because of Cr toxicity [50,51]. It was also reported that Cr(VI) altered chloroplast and membrane structures [26]. Moreover, increasing concentrations of Cr have inhibited the Fe incorporation of protoporphyrin, which is a precursor of the chlorophyll molecule [52,53].

### 4.2. Chlorophyll Pigments, Water Status, and Photosynthetic Activity

The most significant consequence of heavy metal excess is the damage of the photosynthetic apparatus [54]. As with other heavy metals, Cr may affect the most important components participating in photosynthetic machinery, such as photosynthetic pigments, photosystems, electron transport systems, gas exchange, and enzymes involved in $CO_2$ fixation. Chlorophyll, the active molecule of the chloroplast, has a critical role in photosynthesis. The amount of chlorophyll was significantly affected by Cr(VI) treatment. According to our results, leaf chlorosis appeared first in the own-rooted vines treated with 120 μM Cr(VI), followed by the grafted vines. As shown in Figure 3, Cr(VI) toxicity reduced the foliar contents of total chlorophyll and the CCM-200 chlorophyll index; however, this reduction was at the late stages of the treatment cycle. Compared to traditional, destructive methods, CCM-200-m reading applied as an index for the response of relative Chl content to different types of stresses can be used for rapid and low-cost chlorophyll measurement [55]. Chandra and Kulshreshtha [56] reported that plants exposed to Cr stress showed depleted chlorophyll contents that might be due to disrupted chlorophyll biosynthesis.

Moreover, increased chromium content can cause ultrastructural changes in the chloroplasts, leading to the inhibition of photosynthesis [57].

Decreases in chlorophyll contents under Cr toxicity have been reported in different plants species, such as *Pistia stratiotes* [58], *Citrus limonia*, and *Citrus reshni* [59]. Degradation of δ-aminolevulinate acid dehydratase (ALAD) occurred under Cr(VI) toxicity, leading to a decrease in chlorophyll level [38]. δ-aminoleuvulinate acid dehydratase catalyzes the formation of porphobilinogen (PBG), which is a precursor of chlorophyll biosynthesis, from two δ-aminolevulinic acid (ALA) molecules [60–62]. Moreover, our results showed a significant decrease in leaf iron (Fe) concentration in grapevine plants exposed to Cr(VI). In plants, Fe is a component of the photosystems and, thus, is essential for photosynthesis. On the other hand, it is worth noting that Fe deficiency causes a typical leaf chlorosis and toxicity. The visible toxic symptoms of Cr(VI) in our vines were somewhat like those of Fe deficiency. This was partly related to the ability of Cr to displace Fe at physiologically active sites, resulting in leaf toxicity symptoms [63].

As shown in Figure 3, Cr(VI) treatment progressively reduced photosynthesis, and this was coincident with a decline in leaf chlorophyll content. It is widely recognized that the efficiency of photosynthesis may be affected by various environmental and genetic factors, such as water potential, restriction of $CO_2$ gas exchange, photosynthetic pigment degradation, ion toxicity, or nutritional imbalance. A gradual reduction in stem water potential was observed in plants treated with Cr(VI); the reduction depended on the duration of the treatment (Figure 5). The highest reduction was observed in vines after 60 d of treatment. Similar significant reductions were found for stomatal conductance and the net $CO_2$ assimilation rate after 30 d of Cr(VI) treatment. Regarding the two rootstocks used in our experiment, 1103P showed higher values of stem water potential. In our experiment, the net photosynthetic rate decreased from 18.1 to 45.19% at day thirty and from 52.4 to 58.33% at day sixty of the experimentation. On the other hand, the strong reduction in photosynthesis for stressed vines may be due to the inhibition of several enzymes related to the photosynthetic apparatus and to the electron transport system (nonstomatal limitations).

Chloroplastic and ultrastructural damages and the disruption of electron transport mechanisms due to Cr-induced changes might be able to reduce photosynthetic efficiency. Similar reasoning regarding photosynthetic inhibition due to Cr stress was reported by Hayat [64]. Ali et al. in 2013 [65] also reported that Cr-induced changes may cause chloroplast damage, abnormal lamellar systems, and ultrastructural changes, which led to

reduced photosynthesis and other assimilatory mechanisms. Probably, electrons yielded from photochemical reactions may not be used for carbon fixation, and the ratio of ribulose biphosphate carboxylase (Rubisco) activity to that of phosphorenol pyruvate carboxylase (PEPC) decreases in Cr-stressed plants [3].

Van Assche and Clijsters in 1990 [66] reported that local substitutions of Mg by heavy metals such as chromium declined the affinity of Rubisco for $CO_2$. A decrease in Rubisco affinity to $CO_2$ compromised the efficiency of photosynthesis by the limitation of electron capture and transport from photosystem II (PSII) to electron acceptors.

Davies et al. in 2002 [67] noticed that Cr inhibited the photosynthetic process by targeting photosystem II (PSII). Hence, chlorophyll fluorescence seems quite a useful tool to study the photosynthetic apparatus and the action of PSII under heavy metal stress. The ratio of Fv/Fm is an indicator of the efficiency of the photosynthetic apparatus and, therefore, for the photosynthetic strength of the plant. Values of the Fv/Fm ratio >0.8 indicate that plants are healthy and not suffering from photosynthetic stress [68]. Table 4 shows the Fv/Fm ratios of vines following an appropriate period of dark adaptation. It was found that Cr(VI)-stressed vines had lower Fv/Fm ratios compared to the control. However, it was observed that, after thirty days of Cr(VI) treatment, the Fv/Fm ratio remained relatively high in both control and Cr-stressed vines (0.718–0.837). During the following period, especially at the end of the experimental cycle (60 d), strong significant differences were found between the stressed and control vines. As regards the vine cultivars and rootstocks used in the experiment, our results indicated that sixty days after the Cr(VI) treatment the Cabernet Franc vine grafted to the 101-14 Mgt rootstock recorded the lowest value of Fv/Fm ratio (0.494). At this stage, general chlorotic symptoms appeared in most leaves. It should be noted that similar symptoms did not appear before forty-five days of the Cr(VI) treatment. On the other hand, we saw that the net $CO_2$ assimilation rates and stomatal conductance declined in an earlier stage than the Fv/Fm ratio decrease. Subrahmanyam [69] reported that gas exchange parameters were much more sensitive to Cr(VI) treatment than photosystem II efficiency.

There is another mechanism that may be involved in Cr stress tolerance: it was reported that Cr toxicity was related to the elevated response of reactive oxygen species (ROS), which led to oxidative stress and the disorganization of lipids in the cell membrane [70]. The increased generation of toxic reactive oxygen species (ROS) in chloroplasts leads to oxidative damage under abiotic stress [71], inducing the inhibition of electron capture and transport from photosystem II (PII). According to our results presented in Figure 4, chromium toxicity enhanced leaf phenolic substances measured sixty days after the beginning of the treatment. Phenolic compounds protect plants from physiological stresses, such as oxidative stress, by preventing the breakdown of macromolecules and cellular walls [72]. Kisa et al. [73] also found that heavy metal stress caused an increase in the total content of phenolic compounds in corn leaves. In the present work, increased leaf phenolic substances were measured in the stressed vines. Moreover, vines grafted on 101-14 Mgt rootstocks, like own-rooted vines, accumulated increased amounts of phenolic substances in their leaves.

## 5. Conclusions

In conclusion, the obtained results demonstrated that treatment with 120 μM Cr(VI) in a hydroponic culture for sixty days had adverse effects on vine growth and development. Our evidence suggested that: (1) leaf interveinal chlorosis appeared after forty-five days of treatment, as well as overall leaf chlorosis and brown root coloration after sixty days; (2) chromium(VI) treatment significantly increased the total chromium (Cr) concentrations in all parts of the vines; (3) the Cr accumulation in different parts of the vines followed a descending order of roots > trunks > leaves > shoots; (4) Cr toxicity also affected the nutrient element concentrations in different parts of the vines; (5) grafted vines accumulated less Cr in their roots and shoot tissues compared to nongrafted ones; and (6) Cr(VI) stress reduced the concentrations of the most important nutrient elements, as well as the chlorophyll

content, stem water potential, and photosynthetic activity in both cultivars. On the contrary, the total phenolic substances increased under Cr toxicity.

The two rootstocks used in the present work differentiated in Cr uptake and distribution in different parts of the vines, with 1103P being a better excluder for chromium. Thus, in the case of soil with increased chromium concentrations, it is recommended to use the 1103P rootstock. Stressed vines grafted onto this rootstock showed increased vigour, higher leaf chlorophyll contents, and better water relations. It is expected that the data collected and presented in the current manuscript can contribute towards a more thorough knowledge and understanding of Cr toxicity on vine nutrition and physiology, especially in the case of vines that are grown in industrial areas. However, additional research under field conditions is required to further verify all these conclusions and hypotheses caused by Cr stress.

**Author Contributions:** Conceptualization, K.-E.N. and E.Z.; data collection, K.-E.N., S.T. and T.C; data analysis, K.-E.N.; writing and editing, K.-E.N.; nutrient element analysis, K.-E.N. and T.C.; review, S.K. and A.A. All authors have read and agreed to the published version of the manuscript.

**Funding:** This research was funded by the Hellenic Foundation for Research and Innovation (H.F.R.I.), funding number: T1EΔK-04363.

**Institutional Review Board Statement:** Not applicable.

**Informed Consent Statement:** Not applicable.

**Data Availability Statement:** All related data are within the manuscript.

**Acknowledgments:** We would like to thank Areti Bountla for her assistance in lab analyses.

**Conflicts of Interest:** The authors declare no conflict of interest.

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
