# Peer review of "Effects of Chromium Toxicity on Physiological Performance and Nutrient Uptake in Two Grapevine Cultivars (Vitis vinifera L.) Growing on Own Roots or Grafted onto Different Rootstocks"

_horticulturae, doi:10.3390/horticulturae8060493_

Round 1

Reviewer 1 Report

The experimental work refers some effects of Chromium toxicity  on grapevines growing in hydroponic conditions on different scion rootstocks combinations.

During the experiment, lasted 60 days,  nutrient concentration; growth, CO2 assimilation; stomatal conductance and chlorophyll fluorescence were measured.

Results were discussed according to the updated knowledgement on chromium stress physiology.

Author Response

The manuscript has been revised according to reviewers’ comments.

Reviewer 2 Report

The authors in this manuscript study the effects of Cr (VI) toxicity on the growth and physiological performance of the grapevine. They used a hydroponic vine growing experiment, two grape vine cultivars (Merlot and Cabernet Franc) grafted onto two different rootstocks (1103P and 101-14 Mg) were evaluated.

The topic is revealing and deals with an interesting issue in Viticulture management and sustainable viticulture. The introduction is clear and correct, aims are well described. However, in my opinion, the manuscript contains some flaws.In first place, the Introduction section seems too long and many parts could shortened going directly to the manuscript end points as per its title, focused more on Viticulture and more bibliography on this aspect.  I would like to see information about the Cr toxicity in Viticulture, nutritional changes, Cr accumulation in grapevine (Introduction and Discussion section).  

The experimental part and the derived comments should be better assessed and commented. It seems not clear how the Authors derived their conclusions: please give more details and explanation

The conclusion section should be better addressing the manuscript end points outlining the added value of the study. Please add some comments and mention the advantages and limits of the proposed study for Soil management in vineyards.

I recommend the acceptance of this manuscript for publication after considering minor revision. The authors should add in-depth Discussion furthermore and please add suggested comments.

Author Response

I have revised the updated manuscript version according to your comments.

-However, in my opinion, the manuscript contains some flaws. In first place, the Introduction section seems too long and many parts could shortened going directly to the manuscript end points as per its title, focused more on Viticulture and more bibliography on this aspect. 

Concerning the introduction section, I have shortened the introduction. Some very general references have been removed ( 3,4,7).

I would like to see information about the Cr toxicity in Viticulture, nutritional changes, Cr accumulation in grapevine (Introduction and Discussion section).  

It is important to note that there are no appropriate references on grapevine Chromium toxicity.

However, it is interesting to investigate the behaviour of important grapevine rootstocks in Cr stressful conditions.

An additional reference has been added about the Chromium toxicity in grapevine.

- The conclusion section should be better addressing the manuscript end points outlining the added value of the study. Please add some comments and mention the advantages and limits of the proposed study for Soil management in vineyards.

The authors should add in-depth Discussion furthermore and please add suggested comments.

According to your comment, I have totally revised conclusion section. In the revised version they are clearly cited the manuscript end points, outlining the added value of this study.

  1. Leaf interveinal chlorosis appeared after forty-five days of treatment and an overall leaf chlorosis and brown root coloration, after sixty days of treatment.
  2. Chromium (VI) treatment significantly increased the total Chromium (Cr) concentrations, in all parts of the vives.
  3. The Cr accumulation in different parts of vine followed the following descending order: roots > trunks > leaves >shoots.
  4. Cr toxicity also affected the nutrient element concentration in different parts of the vines.
  5. Grafted vines accumulated less Cr in their roots and shoot tissues, when compared to the non-grafted ones.
  6. Cr (VI) stress reduced the concentration of the most important nutrient element as well as the chlorophyll content, stem water potential and photosynthetic activity in both cultivars.
  7. In the contrary, total phenolic substances increased under Cr toxicity.
  8. The two rootstocks used in the present work, differentiated in Cr uptake and distribution in different parts of the vine, with 1103 P being better excluder for Chromium element.
  9. In the case of soils with increased chromium concentrations, it is recommended to use the 1103 P rootstock.
  10. Stressed vines grafted onto this rootstock showed increased vigour, higher leaf chlorophyll content and better water relations.

The current study will contribute towards a more thorough knowledge and understanding of Cr toxicity on vine nutrition and physiology, especially in the case of vines that are grown in industrial areas.

Reviewer 3 Report

In this manuscript the addition of chromium solution to the grapevine potted plants (own-rooted or grafted to 1103P and 101-14 Mgt) on nutrient uptake and physiology of two grapevine cultivars are investigated. This adverse effect of chromium (from the soil or water) on the plant growth and physiology is well known and already investigated on other agricultural crops (and the results obtained in this study are expected and in accordance to the previously published literature), so this manuscript adds very little novelty to the current knowledge and consequently does not deserve to be published in a Horticulturae journal, but may be eventually suitable for some professional journal (after the appropriate revision).

Beside this major flaw, the standard of the manuscript is quite low and several other aspects should be addressed:

  • stem water potential should be written instead of steam (throughout the manuscript),
  • the introduction is too general, and too basic (written as a book),
  • the year when the experiment was conducted is not mentioned,
  • authors do not explain what they mean by ‘suitable concentration of chromium’,
  • latin names of rootstocks are not written with all capital letters,
  • it is not mentioned at which stages/dates was measured the stem water potential,
  • it is not mentioned when the harvest was done,
  • it would be more appropriate that the P values of the statistics are shown in the tables (even for the interactions) and that the different letters are used to separate the significant differences between the mean values,
  • the measuring units (%, mg/kg, etc.) shoul be indicated inside the tables (and not in the title/heading of the table),
  • the majority references used are quite old, confirming that the thematic of the manuscript is not up to date,
  • several minor spelling/grammar errors should be corrected in the text.

Author Response

  • stem water potential should be written instead of steam (throughout the manuscript).

          It has been corrected, as you have mentioned.

  • the introduction is too general, and too basic (written as a book).

         I have shortened the introduction section.

         Some very general references have been removed ( references 3,4,7).

  • the year when the experiment was conducted is not mentioned.

           It has been mentioned the year of experimentation.

  • authors do not explain what they mean by ‘suitable concentration of chromium’.

            It has been explained in the text what we mean by the ‘suitable  concentration of chromium’.

           The selection of the above Cr(VI) concentration was based on preliminary experiments using Cr(VI) at a concentration of 0 – 200 μM, in order to determine the suitable Cr(VI) concentration for our experimentation and the corresponding treatment duration.

  • latin names of rootstocks are not written with all capital letters.

          According to your comment, the latin names of rootstocks were revised.

  • it is not mentioned at which stages/dates was measured the stem water potential.

          The stages of the of the stem water potential measurement like the other physiological parameters are mentioned in the text.

  • it is not mentioned when the harvest was done.

         It has been mentioned in the text, according to your comment.

  • it would be more appropriate that the P values of the statistics are shown in the tables (even for the interactions) and that the different letters are used to separate the significant differences between the mean values.

         Different letters have been used in Tables and Figures to indicate significant differences instead LSD.

  • the measuring units (%, mg/kg, etc.) should be indicated inside the tables (and not in the title/heading of the table)

         The measuring units have been indicated inside the Tables.

  • the majority references used are quite old, confirming that the thematic of the manuscript is not up to date.

         It is important to note that there are no appropriate references on grapevine Chromium toxicity. However, it is interesting to investigate the behavior of important grapevine rootstocks in Cr stressful conditions.

        An additional reference has been added about the Chromium toxicity in grapevine.

  • several minor spelling/grammar errors should be corrected in the text.

         Several errors have been corrected in the text.

Round 2

Reviewer 3 Report

Although the manuscript still has a low scientific standard, and I still think that it would be more appropriate to publish it in some technical journal, it has been improved after revision and it may be accepted for the publication in Horticulturae journal.